# Prevalence and clinical characteristics of increased pancreatic enzymes in patients with severe fever with thrombocytopenia syndrome

**Zhongwei Zhang[1☯‡], Xue Hu[2☯‡], Qunqun Jiang[1], Qian Du[1], Jie Liu[1], Mingqi Luo[1], Liping Deng[1]\*, Yong Xiong [1]\***

**1** Department of Infectious Disease, Zhongnan Hospital of Wuhan University, Wuhan, China, **2** Department of Infectious Diseases, Tongji Hospital, Tongji Medical College and State Key Laboratory for Diagnosis and Treatment of Severe Zoonotic Infectious Disease, Huazhong University of Science and Technology, Wuhan, China

☯ These authors contributed equally to this work.
‡ These authors share first authorship on this work.
\* dengliping@whu.edu.cn (LD); yongxiong64@163.com (YX)

**Data Availability Statement:** All relevant data are within the paper and its Supporting Information files.

## Abstract

### Background and aim

The increased pancreatic enzymes have recently been reported in patients with severe fever with thrombocytopenia syndrome (SFTS). However, its significance has not been elucidated clearly. The aim of this study was to explore the prevalence, clinical characteristics of elevated pancreatic enzymes (amylase and lipase) and its association with AP in patients with SFTS.

### Methods

Data of demographics, comorbid conditions, clinical symptoms, laboratory parameters and survival time of patients with SFTS were collected. Patients were assigned into the non-AP and AP groups according to the diagnostic criteria of AP. Patients in the non-AP group were divided into the normal (<upper limit of normal [ULN]), elevated pancreatic enzymes (EPE) (1–3×ULN) and high pancreatic enzymes (HPE) (>3×ULN) groups according to the serum amylase and lipase levels, and then their clinical data were compared.

### Results

A total of 284 patients diagnosed with SFTS were retrospectively enrolled, including 248 patients in the non-AP group and 36 patients in the AP group. Patients in the non-AP group were composed of 48, 116 and 84 patients in the normal, EPE and HPE groups, respectively. Compared with patients in the normal and EPE groups, patients in the HPE group had higher serum levels of laboratory parameters referring to liver, kidney, heart and coagulation system injury, as well as higher viral load. The cumulative survival rate of patients in the HPE group was significantly lower than that of patients in the normal group. In addition, patients in the AP group also had higher serum levels of laboratory variables reflecting liver,

**Funding:** This work was supported by grants from Key Research and Development Program of Hubei Province, China (2020BCB025). The funders had no role in study design, data collection and analysis, decision to publish, or preparation of the manuscript.

heart, coagulation dysfunction and viral load than patients in the HPE group. The cumulative survival rate of patients in the AP group was significantly lower than that of patients in the HPE group.

## Conclusion

The increased pancreatic enzymes are very common in patients with SFTS, but they are not always associated with AP. Though AP accounts for the majority of deaths for patients with elevated pancreatic enzymes, patients with pancreatic enzymes >3×ULN except for AP also have a high in-hospital mortality rate.

### Author summary

We reported that the prevalence of elevated pancreatic enzymes in patients with SFTS, and demonstrated that the elevation of pancreatic enzymes was very common in SFTS, which might not always be associated with AP. Imaging examinations could be necessary for clinicians to confirm the diagnosis of AP when SFTS patient with elevated pancreatic enzymes. AP might account for the majority of deaths of patients with elevated pancreatic enzymes. However, among patients with increased pancreatic enzymes except for AP, patients with pancreatic enzymes >3×ULN also had a high in-hospital mortality rate.

## Introduction

Severe fever with thrombocytopenia syndrome (SFTS) is an emerging zoonosis caused by SFTS virus (SFTSV) that was reported in the rural areas of China since 2009 [1]. Patients with mild SFTS usually present with fever and thrombocytopenia, accompanied by some nonspecific symptoms such as anorexia, dizziness, headache, nausea, vomiting, diarrhea and abdominal pain, while some critically ill patients could develop multiple organs dysfunction syndrome including encephalopathy, cardiac failure, acute respiratory distress syndrome, severe acute pancreatitis, acute kidney injury and disseminated intravascular coagulation, with a mortality rate varying from 2.5% to 30% in different epidemic areas [2–4].

It is known that the two most common etiologies of AP are gallstones and alcoholism [5]. Nevertheless, diverse pathogens, including bacteria, viruses, fungi and parasites, may lead to infectious pancreatitis [6]. The elevated pancreatic enzymes (amylase and lipase) generally suggest pancreatic injury and are used to diagnosing AP, but they also emerge in acute and critical illness [7]. Although AP can occur in patients with SFTS, and may be associated with poor outcomes [8], it is not certain whether this is true for elevated pancreatic enzymes. Comprehending the association between the disease course of SFTS and increased pancreatic enzymes may be helpful for clinicians to discriminate patients at risk of poor outcomes and prescribe early treatment. The aim of this study was to explore the prevalence, clinical characteristics of elevated pancreatic enzymes (amylase and lipase) and its association with AP in patients with SFTS.

## Patients and methods

### Patients

A total of 284 consecutive patients with SFTS admitted to the Department of Infectious Disease, Zhongnan Hospital of Wuhan University between August 2016 and June 2023 were

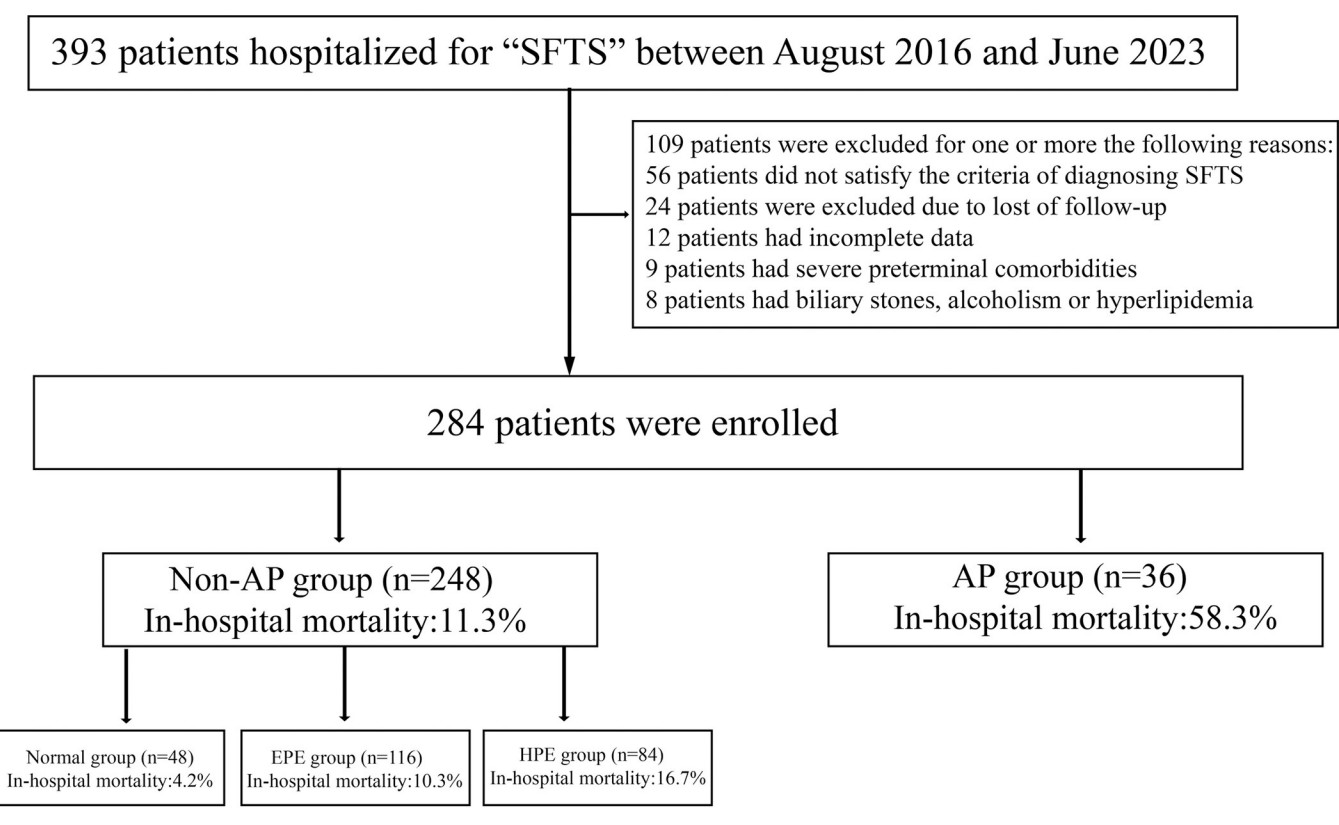

**Fig 1. The study flow chart of the enrollment of patients.**

enrolled into a retrospective cohort. Patients were assigned into the non-AP and AP groups according to the diagnostic criteria of AP. Patients in the non-AP group were divided into the normal (<upper limit of normal [ULN]), elevated pancreatic enzymes (EPE) (1–3×ULN) and high pancreatic enzymes (HPE) (>3×ULN) groups according to the serum amylase and lipase levels (Fig 1).

## Ethics statement

The research involved analysis of anonymised data routinely collected and written informed consent was waived due to the nature of the retrospective study and pandemic nature of the disease. The study was conducted according to the principles expressed in the Declaration of Helsinki and approved by the Ethics Committee of Zhongnan Hospital of Wuhan University (Registration number 2023117K). The results are reported according to the strengthening the reporting of observational studies in epidemiology (STROBE) guidelines.

## Diagnostic criteria

The criteria of diagnosing SFTS were as follows: febrile patients (temperatures of 37.3°C or more for over 24 hours) and decreased platelet count; laboratory-confirmed SFTSV infection by detection of viral RNA in serum via reverse transcriptase polymerase chain reaction.

AP diagnostic criteria were referred to the revision of the Atlanta criteria [9]. Diagnosis of AP met at least two of the following three features: (1) serum amylase and/or lipase levels≥3×ULN; (2) abdominal pain (acute onset of persistent, severe, epigastric pain often radiating to the back); and (3) characteristic findings of AP on imaging examinations. CT was

used for imaging tests. A total of 136 patients were tested for imaging examinations, including 16 patients with abdominal pain and amylase and/or lipase <3×ULN, 31patients with abdominal pain and amylase and/or lipase >3×ULN, 89 patients without abdominal pain but with amylase and/or lipase >3×ULN.

Patients were excluded if they fulfilled one or more of the following reasons: (1) age≤18 years or ≥80 years, (2) the presence of preterminal comorbidities (heart disease New York Heart Association III–IV, severe chronic obstructive pulmonary disease, chronic renal failure), (3) any other types of immunodeficiency, (4) history of malignant tumor, (5) other etiologies for AP, such as biliary stones, alcoholism, hyperlipidemia and autoimmune disease, etc.

During hospitalization, all patients received routine supportive treatment, including bed rest, adequate nutritional support, intensive care and monitoring. Complications including severe coagulation disorders, thrombocytopenia crisis, neutropenia, acute respiratory distress syndrome, acute kidney injury and septic shock were closely monitored and treated immediately.

## Data collection

The medical records of patients with SFTS were reviewed, demographic details, comorbid conditions, symptoms, signs, radiological findings and laboratory tests data including white blood cell (WBC) count and percentage, neutrophils count and percentage, lymphocyte count and percentage, platelet count, hemoglobin, alanine aminotransferase (ALT), aspartate aminotransferase (AST), total bilirubin (TBIL), albumin, globulin, alkaline phosphatase (ALP), gamma glutamyl transpeptidase (GGT), lactate dehydrogenase (LDH), amylase, lipase, blood urea nitrogen (BUN), creatinine, cystatin C, sodium, potassium, calcium, creatinine kinase (CK), creatinine kinase myocardial b fraction (CK-MB), troponin I, brain natriuretic peptide (BNP), prothrombin time (PT), international normalized ratio (INR), prothrombin activity (PTA), activated partial thromboplastin time (APTT), thrombin time (TT), fibrinogen, D-dimer, C-reactive protein (CRP), procalcitonin, interleukin-6 (IL-6), erythrocyte sedimentation rate (ESR), SFTSV viral load, occult blood test (OBT) and survival time were collected.

## Statistical analysis

Categorical variables were shown as numbers (percentages) and were compared by the Chi-square test or Fisher's exact test. Continuous variables were shown as the means ± standard deviations for data with a normal distribution or medians with interquartile ranges (P25-P75) for data with a non-normal distribution, which were compared by the Student's t test or Mann–Whitney U test, respectively. The cumulative survival rates of patients were evaluated using the Kaplan-Meier method and were compared by the Log-rank test. Analysis of receiver operating characteristic (ROC) curves was used to calculate the area under the curve (AUC), and Youden index was used to identify the suggested cutoff value for diagnosing AP in SFTS. All data were analyzed with IBM SPSS statistical analysis software (version 26.0, Chicago, USA), and $P < 0.05$ (two-sided) was considered statistically significant.

## Results

### Demographics, comorbid conditions, clinical manifestations and laboratory tests results of patients classified by pancreatic enzymes levels without AP

A total of 284 consecutive patients diagnosed with SFTS were reviewed, including 248 patients in the non-AP group and 36 patients in the AP group. Patients in the non-AP group were

**Table 1. Comparison of demographics, comorbid conditions and clinical symptoms of patients classified by pancreatic enzymes levels without AP.**

| | Normal group (n = 48) | EPE group (n = 116) | HPE group (n = 84) |
|---|---|---|---|
| Male, n (%) | 26(54.2) | 62(53.4) | 43(51.2) |
| Age (years) | 62±11 | 64±9 | 65±7 |
| Diabetes mellitus, n (%) | 4(8.3) | 4(3.4) | 11(13.1) [#] |
| Hypertension, n (%) | 12(25.0) | 23(19.8) | 25(29.8) [#] |
| Days from onset to admission | 6(4–7) | 7(5–8) | 6(5–7) |
| Clinical manifestations, n (%) | | | |
| Fever >38˚C | 10(20.8) | 26(22.4) | 27(32.1) |
| Headache | 6(12.5) | 16(13.8) | 19(22.6) |
| Dizziness | 12(25.0) | 30(25.9) | 38(45.2) [&##] |
| Cough | 8(16.6) | 28(24.1) | 22(26.2) |
| Sputum | 5(10.4) | 26(22.4) | 14(16.7) |
| Chest distress | 11(22.9) | 16(13.8) | 16(19.0) |
| Anorexia | 31(64.6) | 79(68.1) | 58(69.0) |
| Nausea | 32(66.7) | 75(64.7) | 55(65.5) |
| Vomiting | 9(18.8) | 34(29.3) | 33(39.3) [&] |
| Abdominal pain | 4(8.3) | 12(10.3) | 31(25.8) [&&&###] |
| Diarrhea | 7(14.6) | 22(19.0) | 26(31.0)) [&#] |
| Petechia | 2(4.2) | 10(8.8) | 11(13.1) |
| Consciousness disorder | 4(8.3) | 13(11.2) | 14(17.9) |
| Hepatosplenomegaly | 3(6.3) | 6(5.2) | 9(10.7) |

Note

[&] P value<0.05

[&&] P value<0.01 for comparisons between SFTS patients in the normal and HPE groups.

[#] P value<0.05 for comparisons between SFTS patients in the EPE and HPE groups.

composed of 48, 116 and 84 patients in the normal, EPE and HPE groups, respectively. Compared with patients in the normal group, patients in the HPE group had more presence of vomiting. Patients in the HPE group had higher frequencies of having diabetes mellitus and hypertension than patients in the EPE group. Additionally, dizziness, abdominal pain and diarrhea were significantly overexpressed in the HPE group compared with those in the normal and EPE groups. However, there was no significant difference in age, days from onset to admission, frequencies of fever >38˚C, headache, cough, sputum, chest distress, anorexia, nausea, petechia, consciousness disorder and hepatosplenomegaly between the three groups (Table 1).

Among these laboratory parameters, serum fibrinogen levels of patients in the HPE group were identified to be significantly lower than those of patients in the normal and EPE groups, while ALT, AST, ALP, LDH, creatinine, cystatin C, CK, troponin I, APTT, D-dimer, procalcitonin and IL-6 levels were significantly higher in the HPE group. Higher SFTSV viral load was also detected in the HEP group than in the normal and EPE groups (Table 2).

## Demographics, comorbid conditions, clinical manifestations and laboratory tests results of SFTS patients in the HPE and AP groups

A total of 36 patients were diagnosed with AP, and their serum levels of amylase and/or lipase were all higher than 3×ULN. Their diagnosis patterns of AP were as follows: five patients in line with 1+2 criteria, three patients in line with 1+3 criteria, 28 patients in line with 1+2+3

**Table 2. Comparison of laboratory parameters of patients classified by pancreatic enzymes levels without AP.**

| Variables | Normal range | Normal group (n = 48) | EPE group (n = 116) | HPE group (n = 84) |
|---|---|---|---|---|
| WBC ($10^9$/L) | 3.5–9.5 | 3.4(1.7–5.1) | 3.7(2.3–6.8) | 4.4(2.1–7.6) [&] |
| Neutrophils ($10^9$/L) | 1.8–6.3 | 1.9(1.0–3.0) | 2.0(1.2–5.3) | 3.0(1.1–6.0) [&] |
| Neutrophils (%) | 40.0–75.0 | 66.8(51.1–80.6) | 68.5(50.3–80.4) | 72.1(59.1–84.1) |
| Lymphocyte ($10^9$/L) | 1.1–3.2 | 0.6(0.4–1.3) | 0.8(0.5–1.2) | 0.7(0.5–1.1) |
| Lymphocyte (%) | 20.0–50.0 | 23.7(13.5–33.7) | 21.5(12.2–35.5) | 19.8(10.2–31.9) |
| Platelet ($10^9$/L) | 125–350 | 47(34–65) | 43(29–64) | 42(29–56) |
| Hemoglobin (g/L) | 130–175 | 122±20 | 121±18 | 119±20 |
| ALT (U/L) | 9–50 | 51(31–91) | 66(47–110) [*] | 81(46–140) [&&] |
| AST (U/L) | 15–40 | 103(58–185) | 148(66–289) | 217(91–467) [&&#] |
| TBIL(μmol/L) | 5–21 | 10.7(7.9–15.2) | 11.0(8.6–15.9) | 12.7(9.1–20.0) |
| Albumin (g/L) | 40–55 | 31.1±5.2 | 30.3±4.2 | 29.4±3.8 |
| Globulin (g/L) | 20–40 | 25.6±3.7 | 26.4±4.7 | 27.1±6.0 |
| ALP (U/L) | 30–120 | 63(53–78) | 70(54–92) | 81(62–107) [&&#] |
| GGT (U/L) | 8–57 | 34(49–52) | 38(22–88) | 40(25–90) [&] |
| LDH (U/L) | 125–243 | 449(275–801) | 555(337–843) | 789(481–1072) [&&&###] |
| BUN (mmol/L) | 2.8–7.6 | 5.5(3.8–6.9) | 5.0(3.4–6.6) | 5.5(4.5–8.1) [#] |
| Creatinine (μmol/L) | 64–104 | 70(62–88) | 67(57–88) | 76(66–113) [&##] |
| Cystatin C (mg/L) | 0–1.2 | 1.15(0.90–1.41) | 1.14(0.87–1.42) | 1.34(1.02–1.72) [&&##] |
| Sodium (mmol/L) | 137.0–147.0 | 136.1±4.8 | 135.2±5.7 | 134.2±4.5 |
| Potassium (mmol/L) | 3.5–5.3 | 3.5±0.4 | 3.7±0.6 | 3.8±0.7 [&] |
| Calcium (mmol/L) | 2.11–2.52 | 1.96±0.14 | 1.93±0.16 | 1.92±0.11 |
| CK (U/L) | 0–171 | 223(80–551) | 237(88–682) | 437(182–1215) [&#] |
| CK-MB (U/L) | 0–25 | 25(12–43) | 25(14–42) | 31(19–47) |
| Troponin I (pg/mL) | 0–26.2 | 45.8(20.3–128.0) | 82.2(40.0–152.6) [*] | 115.7(43.0–319.5) [&&#] |
| BNP (pg/mL) | 0–100 | 33.3(15.6–112.3) | 61.6(23.9–162.2) | 85.0(24.0–166.0) [&] |
| PT (s) | 9.4–12.5 | 11.3(10.6–12.1) | 11.2(10.5–11.9) | 11.4(10.8–12.2) |
| INR | 0.85–1.15 | 1.04(0.97–1.11) | 1.03(0.97–1.09) | 1.04(0.99–1.12) |
| PTA (%) | 80–130 | 99(85–109) | 100(90–114) | 97(86–110) |
| APTT(s) | 25.1–36.5 | 39.8(33.6–43.0) | 38.2(32.7–44.4) | 42.8(36.2–51.6) [&##] |
| TT(s) | 10.3–16.6 | 16.6(15.2–18.6) | 17.2(15.9–20.0) | 18.1(16.1–20.1) [&&] |
| Fibrinogen (mg/dL) | 238–498 | 259(210–306) | 260(209–297) | 225(181–291) [&#] |
| D-dimer (ng/mL) | 0–500 | 665(328–1752) | 857(392–1681) | 1315(641–2803) [&&##] |
| CRP (mg/L) | 0–10.0 | 5.1(2.5–9.2) | 8.5(4.5–17.3) [*] | 8.6(3.8–12.1) [&] |
| Procalcitonin (ng/mL) | 0–0.05 | 0.08(0.05–0.36) | 0.16(0.06–0.47) | 0.31(0.14–0.90) [&&##] |
| IL-6 (pg/mL) | 0–7 | 15.0(11.7–30.8) | 19.6(14.1–32.8) | 34.7(23.1–58.6) [&&&###] |
| ESR (mm/h) | 0–20 | 6(4–11) | 7(6–14) | 9(4–12) |
| Viral load ($\log_{10}$ copies/mL) | | 3.4(3.1–3.8) | 3.8(3.4–4.2) [***] | 4.2(3.6–4.5) [&&&##] |
| Urine RBC count | 0–13.1 | 14.5(6.4–47.5) | 17.7(8.5–38.6) | 20.0(10.4–98.0) |
| OBT positivity, n (%) | | 11(22.9) | 18(15.5) | 20(23.8) |

Note

[*] $P$ value<0.05

[***] $P$ value<0.001 for comparisons between SFTS patients in the normal and EPE groups.

[&] $P$ value<0.05

[&&] $P$ value<0.01

[&&&] $P$ value<0.001 for comparisons between SFTS patients in the normal and HPE groups.

[#] $P$ value<0.05

[##] $P$ value<0.01

[###] $P$ value<0.001 for comparisons between SFTS patients in the EPE and HPE groups.

**Table 3. Comparison of demographics, comorbid conditions and clinical symptoms of patients in the HPE and AP groups.**

|  | HPE group (n = 84) | AP group (n = 36) | *P* value |
|---|---|---|---|
| Male, n (%) | 43(51.2) | 12(33.3) | 0.072 |
| Age (years) | 65±7 | 64±10 | 0.548 |
| Diabetes mellitus, n (%) | 11(13.1) | 2(5.6) | 0.223 |
| Hypertension, n (%) | 25(29.8) | 9(25.0) | 0.596 |
| Days from onset to admission | 6(5–7) | 7(5–8) | 0.427 |
| Clinical manifestations, n (%) |  |  |  |
| Fever >38˚C | 27(32.1) | 14(38.9) | 0.475 |
| Headache | 19(22.6) | 11(30.6) | 0.358 |
| Dizziness | 38(45.2) | 14(38.9) | 0.520 |
| Cough | 22(26.2) | 9(25.0) | 0.891 |
| Sputum | 14(16.7) | 5(13.9) | 0.702 |
| Chest distress | 16(19.0) | 12(33.3) | 0.090 |
| Anorexia | 58(69.0) | 29(80.6) | 0.196 |
| Nausea | 55(65.5) | 26(72.2) | 0.470 |
| Vomiting | 33(39.3) | 12(33.3) | 0.537 |
| Diarrhea | 26(31.0) | 12(33.3) | 0.797 |
| Petechia | 11(13.1) | 4(11.1) | 0.763 |
| Consciousness disorder | 14(17.9) | 12(30.6) | 0.042 |
| Hepatosplenomegaly | 9(10.7) | 7(19.4) | 0.197 |

criteria. Compared with patients in the HPE group, patients in the AP group had a higher frequency of consciousness disorder. However, there was no significant difference in age, days from onset to admission, frequencies of having hypertension and diabetes mellitus, presence of cough, sputum, chest distress, anorexia, nausea, headache, dizziness, vomiting, abdominal pain, diarrhea, petechia, hepatosplenomegaly and fever >38˚C (Table 3).

In terms of laboratory findings, fibrinogen levels of patients in the AP group were significantly lower than those of patients in the HPE group, while ALT, AST, ALP, GGT, LDH, amylase, lipase, CK, CK-MB, troponin I, BNP, APTT, TT, CRP, IL-6, procalcitonin, ESR levels and viral load were significantly higher in the AP group than those in the HPE group. No significant differences were observed between the two groups for the remaining laboratory parameters (Table 4).

## Prognosis classification of patients with SFTS

The cumulative survival rates of patients in the normal, EPE and HPE groups were 95.8%, 89.7% and 83.3%, respectively (Fig 2A). The cumulative survival rate of patients in the normal group was significantly lower than that of patients in the HPE group ($P = 0.034$). However, there was no significant difference in the cumulative survival rates between the normal and EPE groups ($P = 0.198$), as well as the same result was observed between the EPE and HPE groups ($P = 0.169$). In addition, the cumulative survival rate of patients in the AP group was significantly lower than that of patients in the HPE group (41.7% vs. 83.3%, $P<0.001$) (Fig 2B). We also demonstrated that AP (OR, 4.183; 95%CI, 1.085–16.121) was an independent risk factor for prognosis of patients with SFTS (S3 Table).

## Diagnosing AP by amylase and lipase in patients with SFTS

Serum amylase and lipase were used to diagnose AP in patients with SFTS. As shown in Fig 3, the AUC of amylase was 0.965 (95% CI 0.938–0.992). With a cutoff value of 260 U/L, the

**Table 4. Comparison of laboratory parameters of SFTS patients in the HPE and AP groups.**

| Variables | Normal range | HPE group (n = 84) | AP group (n = 36) | *P* value |
|---|---|---|---|---|
| WBC ($10^9$/L) | 3.5–9.5 | 4.4(2.1–7.6) | 3.7(2.2–5.9) | 0.364 |
| Neutrophils ($10^9$/L) | 1.8–6.3 | 3.0(1.1–6.0) | 2.1(1.1–4.0) | 0.307 |
| Neutrophils (%) | 40.0–75.0 | 72.1(59.1–84.1) | 65.7(57.6–74.3) | 0.076 |
| Lymphocyte ($10^9$/L) | 1.1–3.2 | 0.7(0.5–1.1) | 0.7(0.5–1.2) | 0.699 |
| Lymphocyte (%) | 20.0–50.0 | 19.8(10.2–31.9) | 24.4(13.8–30.1) | 0.336 |
| Platelet ($10^9$/L) | 125–350 | 42(29–56) | 33(22–44) | 0.070 |
| Hemoglobin (g/L) | 130–175 | 119±20 | 130±21 | 0.262 |
| ALT (U/L) | 9–50 | 81(46–140) | 117(73–256) | 0.007 |
| AST (U/L) | 15–40 | 217(91–467) | 549(263–961) | <0.001 |
| TBIL(μmol/L) | 5–21 | 12.7(9.1–20.0) | 13.4(9.0–18.6) | 0.968 |
| Albumin (g/L) | 40–55 | 29.4±3.8 | 30.2±5.2 | 0.139 |
| Globulin (g/L) | 20–40 | 27.1±6.0 | 29.1±3.2 | 0.138 |
| ALP (U/L) | 30–120 | 40(25–90) | 78(28–288) | 0.040 |
| GGT (U/L) | 8–57 | 81(62–107) | 90(70–209) | 0.042 |
| LDH (U/L) | 125–243 | 789(481–1072) | 1000(702–1653) | 0.047 |
| Amylase (U/L) | 0–90 | 206(173–259) | 418(303–621) | <0.001 |
| Lipase (U/L) | 0–70 | 311(254–406) | 719(455–1037) | <0.001 |
| BUN (mmol/L) | 2.8–7.6 | 5.5(4.5–8.1) | 6.5(4.3–12.1) | 0.323 |
| Creatinine (μmol/L) | 64–104 | 76(66–113) | 98(65–252) | 0.100 |
| Cystatin C (mg/L) | 0–1.2 | 1.34(1.02–1.72) | 1.36(1.14–1.75) | 0.364 |
| Sodium (mmol/L) | 137–147 | 134.2±4.5 | 134.2±6.2 | 0.550 |
| Potassium (mmol/L) | 3.5–5.3 | 3.8±0.7 | 3.9±0.9 | 0.473 |
| Calcium (mmol/L) | 2.11–2.52 | 1.92±0.11 | 1.86±0.12 | 0.090 |
| CK (U/L) | 0–171 | 437(182–1215) | 915(327–1579) | 0.024 |
| CK-MB (U/L) | 0–25 | 31(19–47) | 70(28–125) | 0.001 |
| Troponin I (pg/mL) | 0–26.2 | 115.7(43.0–319.5) | 339.1(102.1–721.1) | 0.001 |
| BNP (pg/mL) | 0–100 | 85(24–166) | 138(62–369) | 0.049 |
| PT (s) | 9.4–12.5 | 11.4(10.8–12.2) | 11.8(10.9–12.8) | 0.187 |
| INR | 0.85–1.15 | 1.04(0.99–1.12) | 1.08(1.00–1.17) | 0.162 |
| PTA (%) | 80–130 | 97(86–110) | 94(83–108) | 0.497 |
| APTT(s) | 25.1–36.5 | 42.8(36.2–51.6) | 51.0(43.0–62.4) | 0.003 |
| TT(s) | 10.3–16.6 | 18.1(16.1–20.1) | 22.1(19.3–27.4) | <0.001 |
| Fibrinogen(mg/dL) | 238–498 | 225(181–291) | 172(152–280) | 0.041 |
| D-dimer (ng/mL) | 0–500 | 1315(641–2803) | 1750(838–3516) | 0.184 |
| CRP (mg/L) | 0–10.0 | 8.6(3.8–12.1) | 11.4(5.2–19.2) | 0.046 |
| Procalcitonin (ng/mL) | 0–0.05 | 0.31(0.14–0.90) | 0.99(0.14–2.34) | 0.029 |
| IL-6 (pg/mL) | 0–7 | 34.7(23.1–58.6) | 47.7(29.2–114.1) | 0.021 |
| ESR (mm/h) | 0–20 | 9(4–12) | 13(7–22) | 0.007 |
| Viral load ($\log_{10}$ copies/mL) |  | 4.2(3.6–4.5) | 5.4(4.3–6.1) | <0.001 |
| Urine RBC counts | 0–13.1 | 20.0(10.4–98.0) | 22.4(10.0–108.4) | 0.813 |
| OBT positivity, n (%) |  | 20(23.8) | 11(30.6) | 0.439 |

sensitivity and specificity of amylase were 94.4% and 91.5%, respectively. The positive predictive value was 61.9%, and the negative predictive value was 99.1%. The AUC of lipase was 0.938 (95% CI 0.896–0.981). With a cutoff value of 406 U/L, the sensitivity and specificity of lipase were 83.3% and 92.3%, respectively. The positive predictive value was 61.2%, and the negative predictive value was 97.4%.

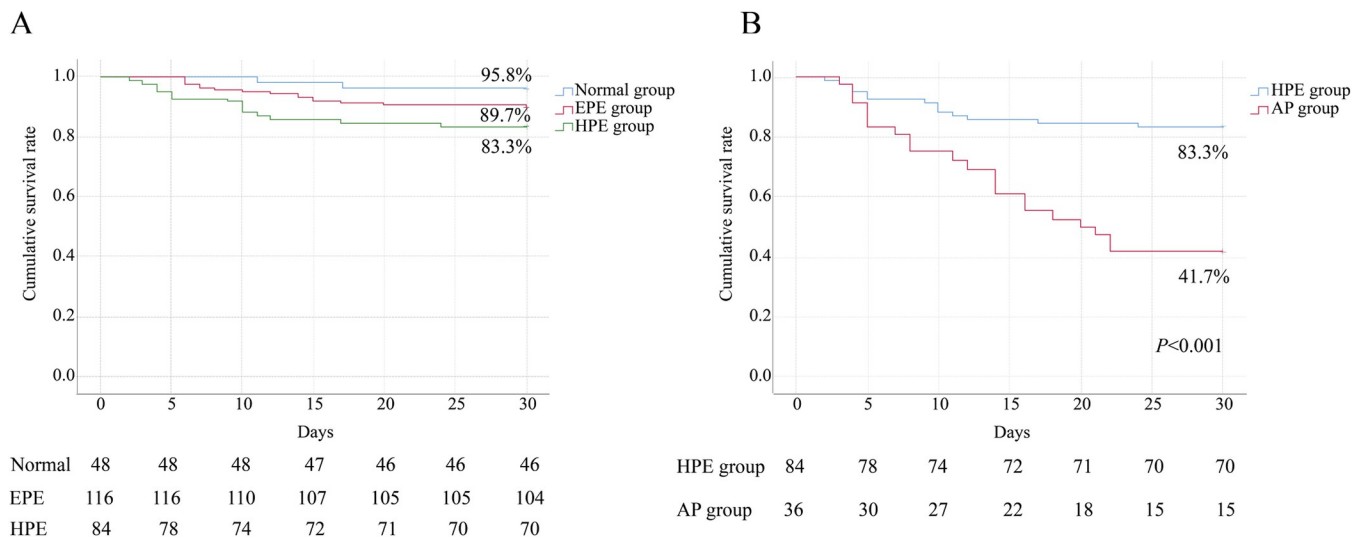

**Fig 2. A, Kaplan–Meier curves showing the cumulative survival rates of patients with SFTS classified by pancreatic enzymes levels without AP.** Comparison of the survival estimates was done using Log-rank test, normal group vs. EPE group, *P* = 0.198; normal group vs. HPE group, *P* = 0.034; EPE group vs. HPE group, *P* = 0.169. B, Kaplan–Meier curves showing the cumulative survival rates of SFTS patients in the HPE and AP groups. Comparison of the survival estimates was done using Log-rank test, HPE group vs. AP group, *P*<0.001.

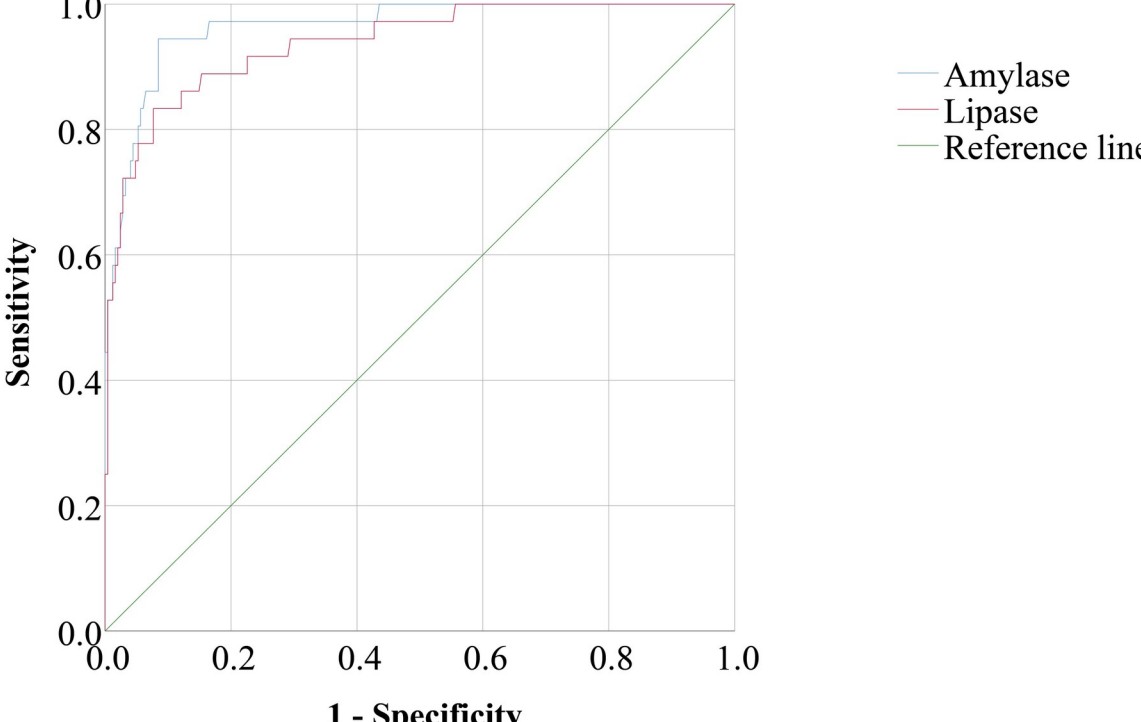

**Fig 3. The area under the receiver operating characteristic curves (AUC) of amylase and lipase for diagnosing AP in patients with SFTS.**

## Discussion

The increased serum pancreatic enzymes including amylase and lipase are common
in critically ill patients without preexisting pancreatic disease [10,11]. It is reported that the prevalence of elevated serum levels of amylase and/or lipase in these patients varies from 14% to 80%, even though they have no prior pancreatic disease [12,13]. The elevation of amylase and lipase concentrations is usually observed in AP, and AP is reported and associated with poor prognosis in patients with SFTS [14]. However, to our knowledge, there is no study exploring the incidence and clinical characteristics of elevated pancreatic enzymes and its association with AP in patients with SFTS.

In this present study, we revealed that the incidence of elevated pancreatic enzymes without AP in SFTS was up to 70.4%, it demonstrated that the elevation of pancreatic enzymes without AP was very common in patients with SFTS. The serum levels of amylase and lipase are the result of a balance between generation and elimination of these enzymes. Amylase is mostly produced and secreted by pancreas and salivary glands, but other organs such as lungs [15–17]. Amylase is removed by the reticuloendothelial system and kidney, and elevated serum levels of amylase can be found in patients with renal insufficiency and renal failure [18]. Lipase is mainly generated and released by the pancreas, which is a key enzyme for the digestion of triglycerides, and is cleared by the kidney [19]. The serum lipase levels raised significantly in patients with chronic renal failure [20]. Gastroenteritis can cause the elevation of pancreatic enzymes, and may be due to the intensified intestinal permeability, which accelerates the reabsorption of amylase and lipase [21]. In this present study, we found that 30.5% of patients with elevated pancreatic enzymes without AP had reduced glomerular filtration rate, and 33.5%, 21.5%, 24.0% of them had vomiting, abdominal pain and diarrhea, respectively. At least in part, it suggested that the elevation of pancreatic enzymes might be explained by the renal insufficiency and intestinal inflammation.

To our knowledge, this is the first time to explore the clinical characteristics of SFTS patients with different serum levels of pancreatic enzymes without AP. Among patients without AP, we reported that patients with pancreatic enzymes >3×ULN had significantly higher serum levels of laboratory parameters referring to liver, kidney, heart, coagulation system injury and viral load than patients with pancreatic enzymes <3×ULN. Furthermore, we also demonstrated that patients with AP had significantly more presence of consciousness disorder, higher serum levels of laboratory variables reflecting liver, heart, coagulation dysfunction and viral load than patients with pancreatic enzymes >3×ULN without AP. The extremely abnormal levels of these laboratory variables generally indicate multiple organs failure, even high mortality [22]. A number of studies have shown that consciousness disorder and viral load are the independent risk factors for prognosis of patients with SFTS [23–25].

It is reported that the elevation of pancreatic enzymes is usually accompanied by shock and mechanical ventilation in critically ill patients in intensive care unit, and prone to multiple organs failure [26]. Some studies have demonstrated that the increased pancreatic enzymes are associated with worse outcomes of patients with coronavirus disease 2019 infection [27,28]. Among SFTS patients without AP, we showed that the cumulative survival rate of patients with pancreatic enzymes >3×ULN was significantly lower than that of patients without elevated pancreatic enzymes. Additionally, we found that the serum levels of amylase and/or lipase of patients diagnosed with AP were all higher than 3×ULN. The cumulative survival rate of patients with AP was significantly lower than that of patients with pancreatic enzymes >3×ULN without AP, and AP was an independent predictor of mortality for patients with SFTS. It suggested that AP might account for the majority of deaths of SFTS people with elevated pancreatic enzymes. To our knowledge, at present, only a study has reported the clinical

features and outcomes of AP in SFTS [8]. However, in the study, only a total of 14 patients have been diagnosed with AP, and the relationship between serum levels of pancreatic enzymes and AP has not be explored. Owing to small number of AP cases, it has not described minutely the impact of AP on outcomes of patients with SFTS and shown that AP is an independent risk factor for prognosis of patients with SFTS.

The cutoff value of amylase and lipase for diagnosing AP in SFTS was identified in our study. It suggested that patients were susceptible to AP when their amylase ≥260 U/L and/or lipase ≥406 U/L. Several studies have revealed that the diagnostic accuracy of pancreatic enzymes is improved significantly when radiological imaging is used to confirm the diagnosis of AP, and imaging can estimate the severity of AP [29–31]. It may be objective and useful for clinicians to identify AP early using imaging examinations when patients with abdominal pain or elevated serum levels of pancreatic enzymes.

The main limitations of our study contain the retrospective study design. In addition, the sample size of this study was relatively small. Therefore, the subsequent analysis of risk factors for AP development in SFTS was not conducted. Finally, because this was a single-center study, the observations made here could not be extrapolated to other centers.

In summary, clinicians should be aware that the elevated pancreatic enzymes are very common in SFTS, and may not always represent a true AP. Imaging examinations could be necessary for clinicians to confirm the diagnosis of AP in SFTS patient with elevated pancreatic enzymes. Though AP may account for the majority of mortality of patients with elevated pancreatic enzymes, patients with pancreatic enzymes >3×ULN except for AP also have a high in-hospital mortality rate.

## Supporting information

**S1 Checklist. STROBE checklist.**
(DOC)

**S1 Table. Comparison of demographics, comorbid conditions and clinical symptoms of SFTS patients in the survivor and non-survivor groups.**
(DOCX)

**S2 Table. Comparison of laboratory parameters of SFTS patients in the survivor and non-survivor groups.**
(DOCX)

**S3 Table. Univariable and multivariable logistic regression analyses of in-hospital mortality of patients with SFTS.**
(DOCX)

**S1 Text. Glossary.**
(DOCX)

## Acknowledgments

The authors wish to thank the patients for participating in this study and all the staff members at our institution.

## Author Contributions

**Conceptualization:** Zhongwei Zhang, Xue Hu, Liping Deng, Yong Xiong.

**Data curation:** Zhongwei Zhang, Xue Hu, Qunqun Jiang, Qian Du, Jie Liu, Mingqi Luo.

**Formal analysis:** Zhongwei Zhang, Xue Hu, Liping Deng, Yong Xiong.

**Funding acquisition:** Yong Xiong.

**Investigation:** Zhongwei Zhang, Xue Hu, Qunqun Jiang, Qian Du, Jie Liu, Mingqi Luo.

**Methodology:** Zhongwei Zhang, Xue Hu, Liping Deng, Yong Xiong.

**Project administration:** Zhongwei Zhang, Xue Hu, Liping Deng, Yong Xiong.

**Resources:** Mingqi Luo, Liping Deng, Yong Xiong.

**Software:** Zhongwei Zhang, Xue Hu.

**Supervision:** Liping Deng, Yong Xiong.

**Validation:** Zhongwei Zhang, Xue Hu.

**Visualization:** Zhongwei Zhang, Xue Hu.

**Writing – original draft:** Zhongwei Zhang, Xue Hu.

**Writing – review & editing:** Liping Deng, Yong Xiong.

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
