## [Decision Letter · Decision Letter 0]

17 Sep 2023

Dear Dr. xiong,

Thank you very much for submitting your manuscript "Prevalence and clinical characteristics of increased pancreatic enzymes in patients with severe fever with thrombocytopenia syndrome" for consideration at PLOS Neglected Tropical Diseases. As with all papers reviewed by the journal, your manuscript was reviewed by members of the editorial board and by several independent reviewers. In light of the reviews (below this email), we would like to invite the resubmission of a significantly-revised version that takes into account the reviewers' comments. 

We cannot make any decision about publication until we have seen the revised manuscript and your response to the reviewers' comments. Your revised manuscript is also likely to be sent to reviewers for further evaluation.

Sincerely,

Wen-Ping Guo

Academic Editor

Elvina Viennet

Section Editor

Reviewer's Responses to Questions

**Key Review Criteria Required for Acceptance?**

**Methods**

-Are the objectives of the study clearly articulated with a clear testable hypothesis stated?

-Is the study design appropriate to address the stated objectives?

-Is the population clearly described and appropriate for the hypothesis being tested?

-Is the sample size sufficient to ensure adequate power to address the hypothesis being tested?

-Were correct statistical analysis used to support conclusions?

-Are there concerns about ethical or regulatory requirements being met?

Reviewer #1: Yes

Reviewer #2: The objectives of the study are clearly articulated. The authors' intention of this study to assess association of pancreatic enzymes with poor outcomes in SFTS. However, they have not excluded patients with acute pancreatitis from this population. The methodology for this paper is otherwise strong and the authors have used appropriate statistical tools where necessary. I have no ethical or regulatory concerns about this paper.

**Results**

-Does the analysis presented match the analysis plan?

-Are the results clearly and completely presented?

-Are the figures (Tables, Images) of sufficient quality for clarity?

Reviewer #1: Yes, but some corrections are needed. The diagnostic criteria for acute pancreatitis in patients with SFTS in the patients in this study should be mentioned more clearly.

Reviewer #2: (No Response)

**Conclusions**

-Are the conclusions supported by the data presented?

-Are the limitations of analysis clearly described?

-Do the authors discuss how these data can be helpful to advance our understanding of the topic under study?

-Is public health relevance addressed?

Reviewer #1: Yes, but some corrections are needed. There are no specific demarcation between the patients who were diagnosed as having acute pancreatitis and those who were diagnosed as not having acute pancreatitis, because imaging tests status are not clearly mentioned in the materials and methods section and the results section.

Reviewer #2: I do not believe that the conclusions support the intent of the paper as stated by the authors. The authors show that elevated pancreatic enzymes in patients with SFTS, particularly when it exceeds > 3 ULN, is associated with poor outcomes. However, we also know from prior studies that acute pancreatitis is an independent risk factor for poor outcomes in SFTS. To truly establish a correlational relationship between elevated enzymes and poor outcomes in SFTS, the authors should have excluded patients with SFTS and acute pancreatitis, as the latter is a major confounder in the analysis of patients with very elevated enzymes. Absent this, this study is not novel and merely restates the association between pancreatitis and poor outcomes. This study in the current form is not enough for physicians to make sense of whether elevated pancreatic enzymes without pancreatitis should raise concerns for poor outcomes.

**Editorial and Data Presentation Modifications?**

Reviewer #1: (No Response)

Reviewer #2: The paper is well presented, I have no major concerns in this regard

**Summary and General Comments**

Reviewer #1: General comments

The authors studied the prevalence of acute pancreatitis (AP) and the outcomes among 284 patients diagnosed as having severe fever with thrombocytopenia syndrome (SFTS). The authors divided the 284 patients into 3 categories (groups) based on the level of amylase and/or lipase, high level (more than 3 times of upper limit of normal (ULN)), middle level (more than the ULN to less than the high level), and normal within the normal level. It is of importance to study the association between clinical course and outcomes of patients with SFTS and the level of pancreatic enzymes. The authors reported that the higher level of pancreatic enzymes, amylase and lipase, in patients with SFTS was closely associated with disease severity and outcomes. Even in the SFTS patients with high level of pancreatic enzymes, the patients with AP had lower survival rate than those without AP.

This reviewer considers that the criteria for diagnosing the patients as having AP is very critical and important for this study. It is written that patients were diagnosed as having AP when the patients had at least 2 of the following 3 criteria, 1) abdominal pain, 2) serum amylase and/or lipase levels (>= 3 times of ULN), and 3) characteristic findings of AP on imaging examinations. However, the AP-diagnostic status for the 284 patients were unclear. This reviewer wishes to know what kind of imaging tests were used, CT, MRI, or others. It should be clarified. This reviewer also wishes to know how many patients were tested for imaging examination. This reviewer considers that it is necessary how many patients were diagnosed as having AP with 1) and 2) criteria, the1) and 3), 2) and 3), and 1), 2) and 3). This reviewer considers that there is no significant difference in the discrimination between the criteria with the combination of 1) and 2) and the combination of 1) and elevated amylase and/or lipase between 1 × ULN and 3 × ULN. 

Specific comments

1) If the line numbers were written, it would be helpful for the reviews to make comments.

2) Page 2, abstract: “then their clinical data were compared” should be “ant then their clinical data were compared”. 

3) Page 2, abstract: The sentence, “Higher viral load was also detected in the HPE group”, does not make sense. “than what”? 

4) Page 3: The description, “an emerging zoonosis infected by SFTS virus (SFTSV)”, does not make sense. A disease (zoonosis) can not be infected with a virus. Human (patient) is infected with a virus.

5) Page 4-5, Diagnostic criteria: Please refer the comments in the General comments.

6) Page 7: The description, “frequency of diabetes ---- ----encephalopathy” needs correction. Symptoms and disease pathophysiology are in the same line. How was the pathophysiology of encephalopathy confirmed in patients with SFTS? It is recommended that “frequency of diabetes” would be replaced by “frequency of having diabetes melitus”.

7) Page 7: The description “38℃” should be “38 ℃”. Please insert a space. This correction is needed in the entire text including tables.

8) Page 9: “difference of other clinical features” should be “difference in other clinical features”?

9) In general, there are several multiple descriptions in the entire text. For instance, the description that the authors reported the association between the disease severity and serum amylase/lipase levels were elucidated for the first time to the authors’ knowledge appeared several times.

10) Discussion section, page 15, last sentence: Is it acceptable to use the term “perfect” for this discussion? This reviewer does not consider so.

Reviewer #2: Although this study is performed well, I have major issues with the lack of confounding as described above. The paper in its current form is not enough to inform clinical practice. The paper will be significantly improved if the analysis is redone without acute pancreatitis patients. This way, one can truly assess the association betweek elevated pancreatic enzymes and clinical outcomes in SFTS.

PLOS authors have the option to publish the peer review history of their article (what does this mean?). If published, this will include your full peer review and any attached files.

Reviewer #1: No

Reviewer #2: No
---

## [Decision Letter · Decision Letter 1]

29 Oct 2023

Dear Dr. xiong,

We are pleased to inform you that your manuscript 'Prevalence and clinical characteristics of increased pancreatic enzymes in patients with severe fever with thrombocytopenia syndrome' has been provisionally accepted for publication in PLOS Neglected Tropical Diseases.

Best regards,

Wen-Ping Guo

Academic Editor

Elvina Viennet

Section Editor

Reviewer's Responses to Questions

**Key Review Criteria Required for Acceptance?**

**Methods**

-Are the objectives of the study clearly articulated with a clear testable hypothesis stated?

-Is the study design appropriate to address the stated objectives?

-Is the population clearly described and appropriate for the hypothesis being tested?

-Is the sample size sufficient to ensure adequate power to address the hypothesis being tested?

-Were correct statistical analysis used to support conclusions?

-Are there concerns about ethical or regulatory requirements being met?

Reviewer #1: This reviewer does have no further comments.

Reviewer #2: (No Response)

**Results**

-Does the analysis presented match the analysis plan?

-Are the results clearly and completely presented?

-Are the figures (Tables, Images) of sufficient quality for clarity?

Reviewer #1: This reviewer does have no further comments.

Reviewer #2: (No Response)

**Conclusions**

-Are the conclusions supported by the data presented?

-Are the limitations of analysis clearly described?

-Do the authors discuss how these data can be helpful to advance our understanding of the topic under study?

-Is public health relevance addressed?

Reviewer #1: This reviewer does have no further comments.

Reviewer #2: (No Response)

**Editorial and Data Presentation Modifications?**

Reviewer #1: (No Response)

Reviewer #2: (No Response)

**Summary and General Comments**

Reviewer #1: This reviewer does have no further comments.

Reviewer #2: Please refer to my review on the previous version of this manuscript for detailed comments. My main criticism of that version was that the authors did not adequately separate the effects of AP in the conclusions they drew from their data. In this revised version, the authors have tried to address this issue as best as they could. I have no further major edits to recommend to this version except a more thorough editorial review to correct minor grammatical errors.

PLOS authors have the option to publish the peer review history of their article (what does this mean?). If published, this will include your full peer review and any attached files.

Reviewer #1: No

Reviewer #2: No

---

## [Editor Report · Acceptance letter]

3 Nov 2023

Dear Dr. Xiong,

We are delighted to inform you that your manuscript, " Prevalence and clinical characteristics of increased pancreatic enzymes in patients with severe fever with thrombocytopenia syndrome ," has been formally accepted for publication in PLOS Neglected Tropical Diseases.

Best regards,

Shaden Kamhawi

co-Editor-in-Chief

Paul Brindley

co-Editor-in-Chief
